

# Intra- and inter-session reliability and repeatability of an infrared thermography device designed for materials to measure skin temperature of the triceps surae muscle tissue of athletes

Cesar Calvo-Lobo[1], Marta San-Antolín[2], Daniel García-García[1,3], Ricardo Becerro-de-Bengoa-Vallejo[1], Marta Elena Losa-Iglesias[4], Julia Cosín-Matamoros[1], Israel Casado-Hernández[1], Eva María Martínez-Jiménez[1], Victoria Mazoteras-Pardo[5] and David Rodríguez-Sanz[1]

[1] Facultad de Enfermería, Fisioterapia y Podología, Universidad Complutense de Madrid, Madrid, Spain
[2] Departamento de Psicología, Universidad de Valladolid, Valladolid, Valladolid, Spain
[3] Escuela de Doctorado, Universidad Europea de Madrid, Villaviciosa de Odón, Madrid, Spain
[4] Facultad de Ciencias de la Salud, Universidad Rey Juan Carlos, Alcorcón, Madrid, Spain
[5] Department of Nursing, Physiotherapy and Occupational Therapy, School of Physiotherapy and Nursing, Universidad de Castilla La Mancha, Toledo, Spain, Spain

Corresponding author
Marta San-Antolín,
marta.sanantolin@uva.es

## ABSTRACT

**Background:** Infrared thermography devices have been commonly applied to measure superficial temperature in structural composites and walls. These tools were cheaper than other thermographic devices used to measure superficial human muscle tissue temperature. In addition, infrared thermography has been previously used to assess skin temperature related to muscle tissue conditions in the triceps surae of athletes. Nevertheless, the reliability and repeatability of an infrared thermography device designed for materials, such as the Manual Infrared Camera PCE-TC 30, have yet to be determined to measure skin temperature of the triceps surae muscle tissue of athletes.

**Objective:** The purpose was to determine the procedure's intra- and inter-session reliability and repeatability to determine skin temperature within the Manual Infrared Camera PCE-TC 30 thermography device in the triceps surae muscle tissue of athletes, which was initially designed to measure the superficial temperature of materials.

**Methods:** A total of 34 triceps surae muscles were bilaterally assessed from 17 healthy athletes using the Manual Infrared Camera PCE-TC 30 thermography device to determine intra- (at the same day separated by 1 h) and inter-session (at alternate days separated by 48 h) reliability and repeatability of the skin temperature of the soleus, medial and lateral gastrocnemius muscles. The triceps surae complex weas measured by a region of interest of 1 cm$^2$ through five infrared thermography images for each muscle. Statistical analyses comprised intraclass correlation coefficient (ICC), standard error of measurement (SEM), minimum detectable change (MCD), systematic error of measurement, correlation ($r$), and Bland-Altman plots completed with linear regression models ($R^2$).

**Results:** Intra- and inter-session measurements of the proposed infrared thermography procedure showed excellent reliability ($ICC_{(1,2)}$ = 0.968–0.977), measurement errors (SEM = 0.186–0.232 °C; MDC = 0.515–0.643 °C), correlations ($r$ = 0.885–0.953), and did not present significant systematic error of measurements ($P > 0.05$). Adequate agreement between each pair of measurement moments was presented by the Bland-Altman plots according to the limits of agreement and non-significant linear regression models ($R^2$ = 0.000–0.019; $P > 0.05$).

**Conclusions:** The proposed procedure to determine skin temperature within the Manual Infrared Camera PCE-TC 30 thermography device presented excellent intra- and inter-session reliability and repeatability in athletes' triceps surae muscle tissue. Future studies should consider the SEM and MDC of this procedure to measure the skin temperature of soleus, medial, and lateral gastrocnemius muscles to promote triceps surae muscle prevention and recovery in athletes.

## HIGHLIGHTS

- Prior high-quality thermography devices were reliable in measuring skin temperature
- The proposed device was initially designed for materials with superficial temperature
- This device is reliable and repeatable for measuring triceps surae skin temperature
- This cheap tool could promote triceps surae prevention and recovery in athletes

## INTRODUCTION

Infrared thermography was proposed as one of the most relevant non-ionizing radiation tools to assess skin temperature related to muscle tissue conditions, such as compartmental syndrome. Nevertheless, there is a lack of scientific evidence for other musculoskeletal conditions (*Sanchis-Sánchez et al., 2014*). Among different application fields, the use of inflammation and perfusion-based conditions may be evaluated by thermographic imaging in sports medicine (*Ramirez-GarciaLuna et al., 2022*). This device has been used to determine the superficial temperature of muscle tissue of lower limbs from athletes for injury prevention (*Côrte et al., 2019*), muscle activity assessment (*Rodriguez-Sanz et al., 2019*), and evaluations before and after treatments (*Rodriguez et al., 2018*; *Benito-de-Pedro et al., 2019*). In addition, assessments before and after running were performed in the triceps surae from athletes as a main focus linked to musculoskeletal conditions (*Rodríguez-Sanz et al., 2017, 2018*; *Rodriguez et al., 2018*; *Benito-de-Pedro et al., 2019*).

Different thermography methods to determine skin temperature in the calves' muscular region showed good correlations among them ($R^2$ = 0.885–0.924) and between both sides ($R^2$ = 0.754–0.881; $r$ = 0.868–0.939) within adequate agreement by Bland-Altman plots (*Ludwig et al., 2014*). Thermography assessment was used to measure the cutaneous temperature of triceps surae muscles in soccer players with equinus condition *vs.* non-equinus condition after running (*Rodriguez-Sanz et al., 2018*), as well as at rest

(*Rodríguez-Sanz et al., 2017*). In addition, this tool was applied to determine the treatment effects after different physical therapy interventions, such as dry needling and ischemic compression, in triathletes (*Benito-de-Pedro et al., 2019*). In addition, infrared thermography was utilized to measure skin temperature changes after compressive *vs.* standard stockings use in athletes (*Rodriguez et al., 2018*). All these evaluations were carried out by the FLIR/SC3000/QWIP Thermacan thermographic tool to measure the skin temperature of the triceps surae muscle tissue.

The FLIR/SC3000/QWIP Thermacan infrared thermal device presented an 8–9 μm spectral range, a temperature sensitivity of 0.02 K, a display of 320 × 240 pixels with 20° lens and a spatial resolution of 1.1 mrad, being frequently used to evaluate the superficial temperature in the human tissue with adequate reliability and repeatability (*Rodríguez-Sanz et al., 2017*, *2018*; *Rodriguez et al., 2018*; *Benito-de-Pedro et al., 2019*), by both manual and automatic thermographic software package measurement methods with an adequate agreement and excellent intraclass correlation coefficient (ICC > 0.80) (*Requena-Bueno et al., 2020*). Prior statistical procedures, such as ICC and Bland-Altman plots, including limits of agreement (LoA), were used to compare infrared thermographic values in the lower limbs showing that both manual and automatic definition devices presented an excellent ICC from 0.92 to 0.99 with an adequate agreement by visual distribution and similar LoA by the ThermoHuman® devices (*Fernandez-Cuevas et al., 2017*; *Requena-Bueno et al., 2020*), excellent inter-session reproducibility with an ICC of 0.88 using the digital infrared camera IRTIS-2000® (*Zaproudina et al., 2008*), and almost perfect agreement in replication with an ICC from 0.94 to 0.97 by the Thermofocus® thermal imaging device (*Petrova et al., 2018*).

Nevertheless, these devices were more expensive than an infrared thermography device designed to evaluate superficial temperature on materials. One of these thermographic devices was the Manual Infrared Camera PCE-TC 30. This tool displayed a sensor resolution of 80 × 80, a measurement range from 0 °C to 250 °C, a display of 320 × 240 pixels, a thermal sensitivity of 80 mK, and an 8 mm lens (*De Villoria et al., 2011*; *Pérez-Urrestarazu et al., 2014*). Despite this, the thermal imaging system presented very low parameters concerning geometrical resolution (80 × 80 pixels) and thermal sensitivity of 80 mK, while other thermal imaging systems displayed high definition resolution (1,280 × 960 pixels) and higher thermal sensitivity of 20 mK (*Fernandez-Cuevas et al., 2017*; *Requena-Bueno et al., 2020*), the Manual Infrared Camera PCE-TC 30 geometrical resolution and thermal sensitivity features could present adequate reliability to measure triceps surae muscle tissue temperature variations (*Rodríguez-Sanz et al., 2017*, *2018*; *Rodriguez et al., 2018*; *Benito-de-Pedro et al., 2019*). Indeed, the PCE-TC 30 thermal camera has already been employed to assess the thermal behavior of fencing uniforms in athletes (*Lamberti et al., 2020*), but the reliability and repeatability of this tool directly on the skin of the human muscle have yet to be determined.

This infrared thermography device was used to measure superficial temperature in structural composites and walls according to quality inspections, such as reproducibility, stability, reliability, and operating temperature (*De Villoria et al., 2011*; *Pérez-Urrestarazu et al., 2014*). Although this tool is used to assess thermal temperature in the fencing

uniforms of athletes (*Lamberti et al., 2020*), it has not been applied directly to measure superficial human muscle tissue temperature, which could be of interest in the triceps surae of athletes (*Rodríguez-Sanz et al., 2017*, *2018*; *Rodriguez et al., 2018*; *Benito-de-Pedro et al., 2019*; *Requena-Bueno et al., 2020*).

We hypothesized that this device—developed initially for non-*Vivo* structures—could provide adequate reliability and repeatability to determine skin temperature in the triceps surae muscle human tissue of athletes, being less expensive than the infrared thermography device used in human studies. Thus, the purpose of the present study was to determine the intra- and inter-session reliability and repeatability of the procedure to assess skin temperature within the Manual Infrared Camera PCE-TC 30 thermography device in the triceps surae muscle tissue of athletes, which was designed initially to measure the superficial temperature of materials.

## MATERIALS AND METHODS

### Study design

The present study was carried out from January 2020 to May 2021 according to The Strengthening the Reporting of Observational Studies in Epidemiology (STROBE) criteria (Appendix 1). The Helsinki Declarations, as well as specific human experimentation ethical rules, were taken into account. The ethics committee for clinical interventions of the Hospital Clínico San Carlos, Madrid (Spain) approved this study with internal code number 20/021-E on January 20, 2020. Before the study began, all participants signed the informed consent form. In addition, the present study was supported by Contract 83 between Complutense University and PCE Ibérica S.L. (Reference number: 6-2020), providing a specific grant for this research project.

### Sample size

The sample size determination was calculated by bi-variate correlations statistical procedures through G*Power 3.1.9.2 program (G*Power©, from Dusseldorf University, in Germany), considering a 0.4 correlation coefficient to achieve a moderate correlation (*Lobo et al., 2016*) between infrared thermography measurements, applying a 1-tailed hypothesis, a 0.05 α error and a 0.80 power. Lastly, the sample size was 34 triceps surae muscles to achieve the required thermography measurements for a 0.801 actual power.

### Participants

Thirty-four triceps surae muscles were bilaterally analyzed from 17 healthy athletes considering a consecutive sampling recruitment procedure. Inclusion criteria comprised specifically healthy athletes aged 18–65 years providing the consent information document previously, carrying out sports activities and training for at least 2 h as well as 1 day per week, with moderate (level-II) or vigorous (level-III) intensities for physical activity with metabolic equivalent indexes greater than 600 METs/min/week, measured by the International Questionnaire for Physical Activity (IPAQ) (*Roman-Viñas et al., 2010*).

Exclusion criteria included muscle soreness, congenital dysfunctions, neuromuscular conditions, rheumatic alterations, body mass index (BMI) greater than 31 kg/m$^2$, previous

neurological conditions, prior surgeries and skin pathologies. Some alterations in the lower limbs region (*i.e.*, chronic ankle instability presence, prior sprains or previous fractures) were also excluded according to the thermographic influence of compartmental, stress, inflammation, and perfusion-based conditions (*Sanchis-Sánchez et al., 2014*; *Ramirez-GarciaLuna et al., 2022*). Lastly, difficulties or inability to carry out the procedure to complete the study course, explained below, were considered as exclusion criteria.

## Procedure

Intra- and inter-session reliability and repeatability of the skin temperature of the triceps surae muscles were bilaterally assessed by the Manual Infrared Camera PCE-TC 30 to determine measurement agreement and concordance at the same day separated by 1 h (considered as intra-day measurements) and alternate days separated by 48 h (considered as inter-day measurements), respectively. Indeed, participants were asked to continue with their daily life and physical exercise routine (avoiding unusual efforts or activity changes) between measurements and not taking prescribed medications at the prior week nor vasomotor substances (*i.e.*, caffeine) on the same measurement day, as well as heavy metals were not allowed. A period of 5 min of acclimatization of the subjects to the room was applied (Fig. 1). All measurements were assessed with patients standing up in a relaxed position in the same room within a 24.1 °C ± 1 °C temperature and a 45% ± 10% humidity, without direct ventilation flow toward examiners or participants (*Rodríguez-Sanz et al., 2017*).

## Descriptive data

Descriptive data including sex (categorized as male or female), age (measured in years), height (measured in cm), weight (measured in kg), and BMI (expressed as $kg/cm^2$ following the Quetelet's index) (*Garrow, 1986*), main sports category (divided into fitness considered as bodybuilding exercise or soccer), side (categorized as right or left), and dominance (expressed as yes or no), and smoker (expressed as yes or no) were detailed (*Calvo-Lobo et al., 2019*). As a tool with adequate psychometric properties, metabolic equivalent index per minute per week (METs/min/week) was evaluated by the IPAQ to determine physical activity and its categorization as moderate (600–1,500 METs/min/week) and vigorous (≥1,500 METs/min/week) physical activity (*Gauthier, Lariviere & Young, 2009*).

## Infrared thermography

We used a Manual Infrared Camera PCE-TC 30 (PCE Instruments UK Ltd, Southampton, United Kingdom), which displayed a sensor resolution of 80 × 80, a measurement range from 0 °C to 250 °C, a display of 230 × 240 pixels, a thermal sensitivity of 80 mK and an 8-mm lens (*De Villoria et al., 2011*; *Pérez-Urrestarazu et al., 2014*). The infrared thermography imaging process was performed with the participant standing up in a relaxed position 1 m from the camera. Bilaterally, the triceps surae complex, including
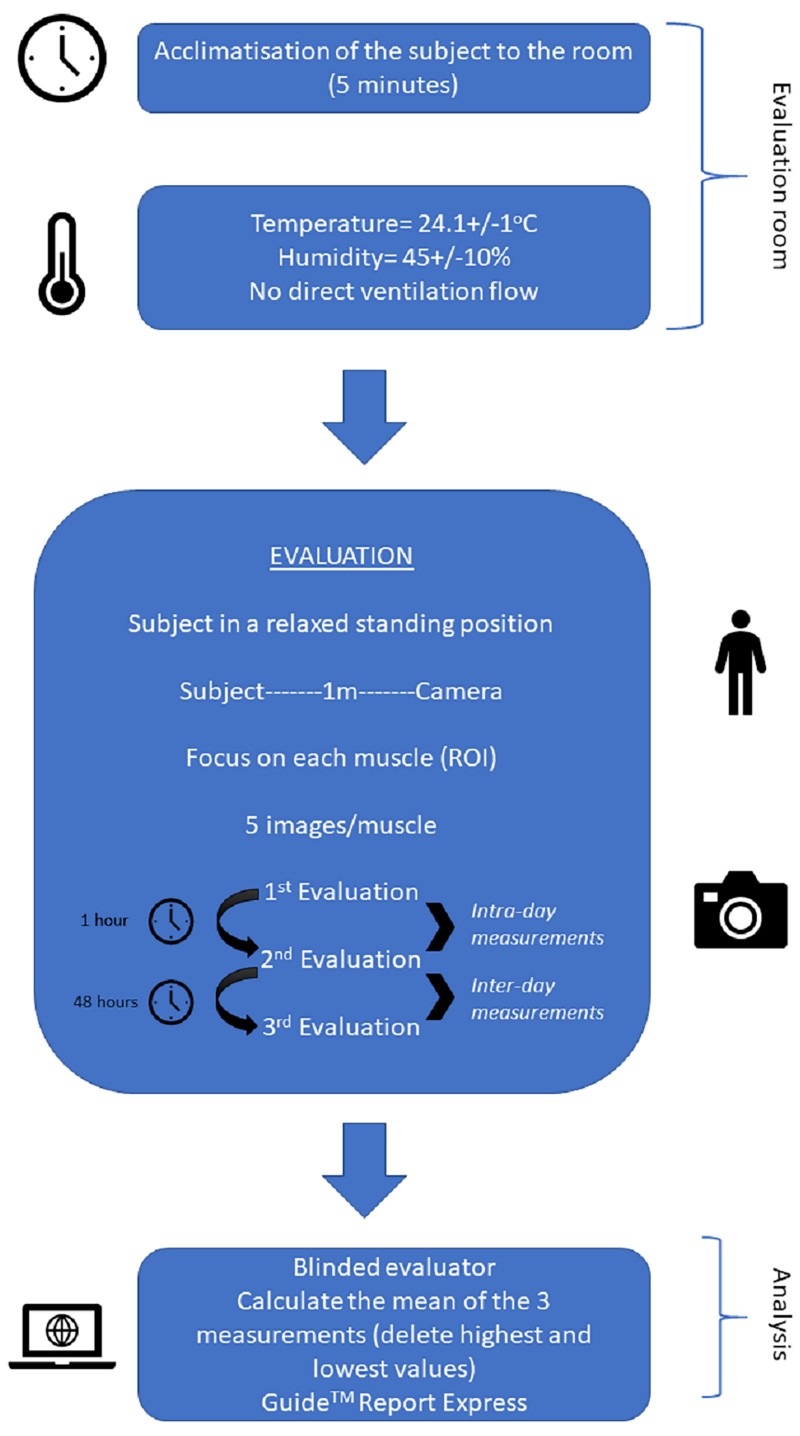

**Figure 1 Infographics for evaluating and analyzing the infrared thermography images of the triceps surae complex.**

lateral (Figs. 1A and 1B) and medial (Figs. 1C and 1D) gastrocnemius, as well as soleus (Figs. 1E and 1F) muscles, was measured by a region of interest (ROI) through five infrared thermography images for each muscle. Removing the highest and lowest values, the mean of the three measurements was used for data analysis.

Infrared images and data were analyzed by a blinded and experienced evaluator using the Guide™ Report Express (PCE Instruments UK Ltd, Southampton, United Kingdom) (*Rodríguez-Sanz et al., 2017*). This software provided the mean thermal value (°C) of the selected ROI of 1 cm$^2$ coinciding with the center of a landmark for each muscle (Fig. 2). These landmarks were used to determine superficial skin temperature and placed superior to the Achilles tendon for the soleus muscle and in the thickest part of the medial and lateral gastrocnemius muscles according to prior similar studies (*Benito-de-Pedro et al., 2019*; *Rojas-Valverde et al., 2021*).

## Statistical analyses

The 24.0 Statistical Package Program for Social Science (named SPSS, from IBM-Corp, in Armonk, NY, USA) was used for data analyses. The α error was set at 0.05, and thus a *P*-value lower than 0.05 was considered statistical significance. The Kolmogorov-Smirnov statistical test and visual inspection of histograms were considered to detail normality distribution. Data adjusted to normal distribution were detailed through means ± standard deviations (SD) in conjunction with the upper and lower limits of 95% confidence interval (CI). Data adjusted to non-normal distribution were detailed through medians ± interquartile ranges (IR). Infrared thermography measurements for intra- and inter-session evaluations were compared through paired-sample Student t-tests considering parametric tests and Wilcoxon tests regarding non-parametric tests. ICC analyzed the reliability and repeatability between each pair of measurements for bidirectional absolute agreement and Pearson (*r*) or Spearman (*ρ*) correlation coefficients as parametric or non-parametric tests, respectively. Furthermore, $ICC_{(2,1)}$ values were specifically interpreted as poor for $<0.40$ $ICC_{(2,1)}$, weak for 0.40–0.59 $ICC_{(2,1)}$, good for 0.60–0.74 $ICC_{(2,1)}$, and excellent for 0.75–1.00 $ICC_{(2,1)}$ (*Calvo-Lobo et al., 2019*).

Next, correlation coefficients were specifically interpreted as weak for 0.00–0.40 *r* or *ρ*, moderate for 0.41–0.69 *r* or *ρ*, and strong for 0.70–1.00 *r* or *ρ* (*Lobo et al., 2016*). Standard errors for measurements (SEM) values were detailed through $SD \times \sqrt{(1 - ICC)}$. After, minimum detectable changes (MDC) values were detailed through $\sqrt{2} \times 1.96 \times SEM$ for 95% CI. Both MDC and SEM were detailed through Bland and Altman recommendations (*Calvo-Lobo et al., 2019*). Limits for agreement (LoA) for each pair of measurements were detailed through differences means $\pm 1.96 \times SD$ for 95% CI in line with Bland and Altman (*Bland & Altman, 2010*; *Calvo-Lobo et al., 2019*).

In addition, Bland-Altman plots were shown to detail visual agreements for each pair of measurements showing systematic measurement errors of the differences in means distributions for each pair of measurements located at the Y-axis with regards to the means for each pair of measurements located at the X-axis. These Bland-Altman plots were shown in conjunction with linear regression models. $R^2$ coefficients were calculated to detail the adjustment quality. The mean values for each pair of measurements were considered independent variables. Lastly, the differences for each pair of measurements were considered dependent variables (*Bland & Altman, 2010*).

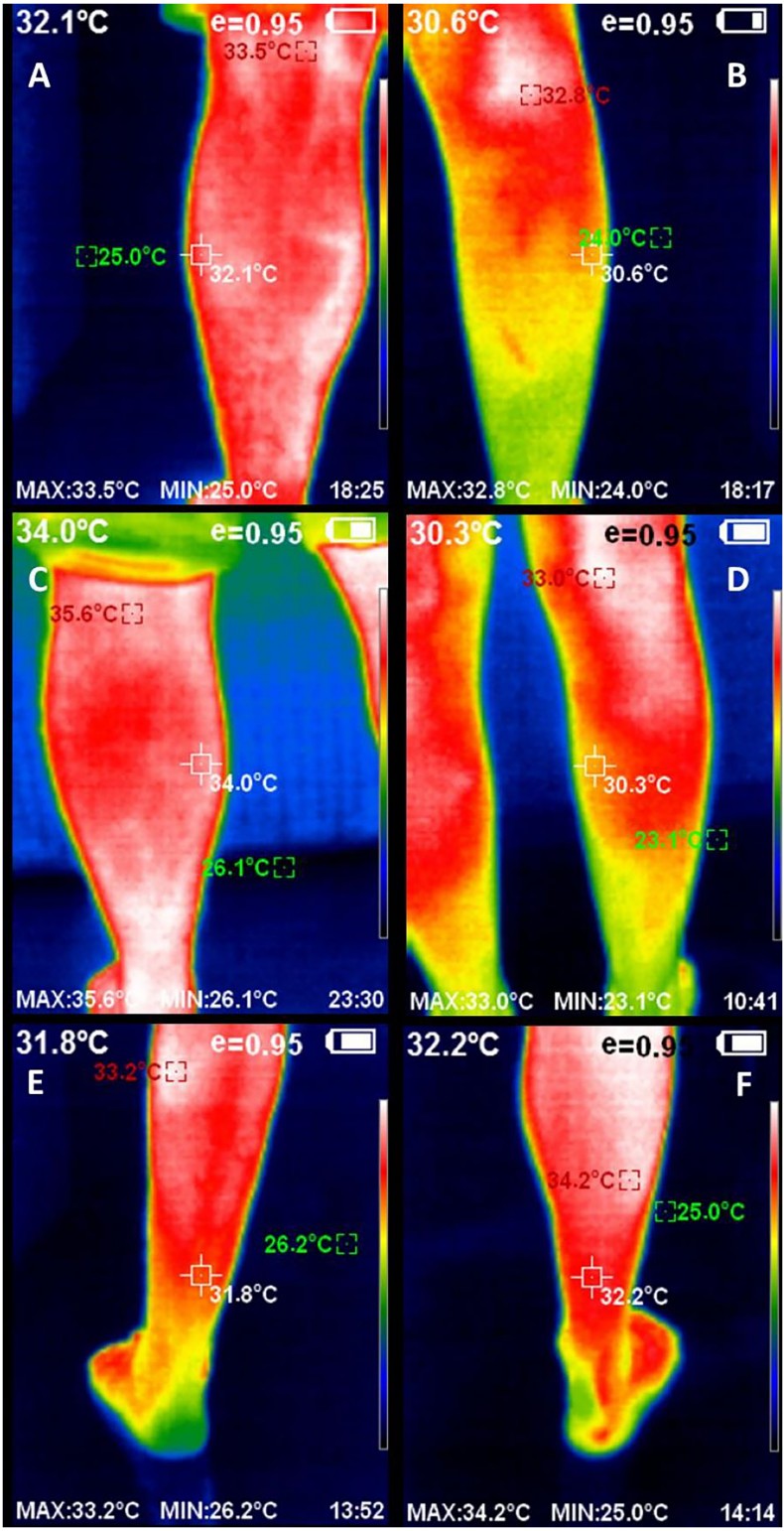

**Figure 2 Infrared thermography images of the triceps surae complex, including left (A) and right (B) lateral gastrocnemius, left (C) and right (D) medial gastrocnemius, and left (E) and right (F) soleus muscles, including the thermal values of the region of interest.**

**Table 1 Descriptive data and normality statistics and significance according to Kolmogorov-Smirnov test.**

| Variables | Descriptive data | K-S statistic | K-S P-value |
|---|---|---|---|
| Age (years) | 41.76 ± 14.42 (36.73–46.79)[‡] | 0.126 | 0.189 |
| Weight (kg) | 68.57 ± 14.57 (63.40–73.64)[‡] | 0.098 | 0.200 |
| Height (m) | 1.69 ±0.09 (1.66–1.73)[‡] | 0.142 | 0.078 |
| BMI (kg/m$^2$) | 23.45 ± 3.47 (22.24–24.66)[‡] | 0.095 | 0.200 |
| IPAQ (METs/min/week) | 3,276.08 ± 1,876.49 (2,621.34–3,930.82)[‡] | 0.110 | 0.200 |
| Soleus at baseline (°C) | 31.31 ± 1.41 (30.82–31.81)[‡] | 0.119 | 0.200 |
| Soleus after 1 h (°C) | 31.47 ± 1.80 (31.34–32.13)[‡] | 0.090 | 0.200 |
| Soleus after 48 h (°C) | 31.55 ± 1.03 (31.18–31.91)[‡] | 0.096 | 0.200 |
| Medial gastrocnemius at baseline (°C) | 31.33 ± 1.40 (30.84–31.82)[‡] | 0.138 | 0.098 |
| Medial gastrocnemius after 1 h (°C) | 31.66 ± 1.81 (31.42–32.22)[†] | 0.155 | 0.039[*] |
| Medial gastrocnemius after 48 h (°C) | 31.55 ± 1.08 (31.17–31.93)[‡] | 0.097 | 0.200 |
| Lateral gastrocnemius at baseline (°C) | 31.24 ± 1.36 (30.76–31.71)[‡] | 0.086 | 0.200 |
| Lateral gastrocnemius after 1 h (°C) | 31.77 ± 1.11 (31.38–32.16)[‡] | 0.146 | 0.065 |
| Lateral gastrocnemius after 48 h (°C) | 31.52 ± 1.08 (31.14–31.90)[‡] | 0.117 | 0.200 |

Notes:
[*] $P < 0.05$ was considered as statistically significant for a 95% CI.
[‡] Mean ± standard deviation and the upper and lower limits of 95% CI were used.
[†] Median ± interquartile range and the upper and lower limits of 95% CI were used.
BMI, body mass index; CI, confidence interval; IPAQ, International Physical Activity Questionnaire; K-S, Kolmogorov-Smirnov test; MET/min/week, metabolic equivalent index per minute per week.

# RESULTS

## Descriptive data

The final sample comprised 34 triceps surae muscles bilaterally from 17 healthy athletes, nine (52.9%) males and eight (47.1%) females, with mean ± SD (95% CI) age of 41.76 ± 14.42 (36.73–46.79) years, the weight of 68.57 ± 14.57 (63.40–73.64) kg, the height of 1.69 ± 0.09 (1.66–1.73) m, and BMI of 23.45 ± 3.47 (22.24–24.66) kg/m$^2$. Regarding the main sports category, these athletes performed in fitness ($n = 14$; 82.40%) and soccer ($n = 3$; 17.6%). All athletes presented the dominant right side ($n = 17$; 100%); most were non-smokers ($n = 12$; 70.60%). Considering the IPAQ, the mean ± SD (95% CI) of metabolic equivalents index per minute per week was 3,276.08 ± 1,876.49 (2,621.34–3,930.82) METs/min/week, including eight (47.10%) athletes who performed vigorous physical activity and nine (52.90%) athletes who performed moderate physical activity. Table 1 shows the normality statistics and significance according to the Kolmogorov-Smirnov test.

## Intra-session reliability and repeatability

According to Table 2, intra-session measurements of the infrared thermography device designed for materials (Manual Infrared Camera PCE-TC 30) in the triceps surae muscle tissue of athletes showed excellent reliability ($ICC_{(1,2)} = 0.969$–0.977), measurement errors (SEM = 0.186–0.212 °C; MDC = 0.515–0.587 °C) and did not present any statistically significant systematic error of measurements ($P > 0.05$).

**Table 2 Intra-session reliability and repeatability of an infrared thermography device designed for materials (Manual Infrared Camera PCE-TC 30) in athletes' triceps surae muscle tissue.**

| Infrared thermography (°C) | Baseline measurements (95% CI) | After 1 h measurements (95% CI) | ICC$_{(1,2)}$ (95% CI) | SEM | MDC | P-value* |
|---|---|---|---|---|---|---|
| Soleus | 31.31 ± 1.41 [30.82–31.81] | 31.33 ± 1.40 [30.84–31.82] | 0.977 [0.954–0.988] | 0.212 | 0.587 | 0.822[‡] |
| Medial gastrocnemius | 31.47 ± 1.80 [31.34–32.13] | 31.66 ± 1.81 [31.42–32.22] | 0.966 [0.931–0.983] | 0.208 | 0.576 | 0.467[†] |
| Lateral gastrocnemius | 31.55 ± 1.03 [31.18–31.91] | 31.55 ± 1.08 [31.17–31.93] | 0.969 [0.938–0.984] | 0.186 | 0.515 | 0.982[‡] |

Notes:
[‡] Mean ± standard deviation and Student t test for paired samples were used.
[†] Median ± interquartile range and Wilcoxon test for paired samples were used.
[*] $P < 0.05$ was considered as statistically significant for a 95% CI.
CI, confidence interval; ICC, intraclass correlation coefficient; MDC, minimum detectable change; SEM, standard error of measurement.

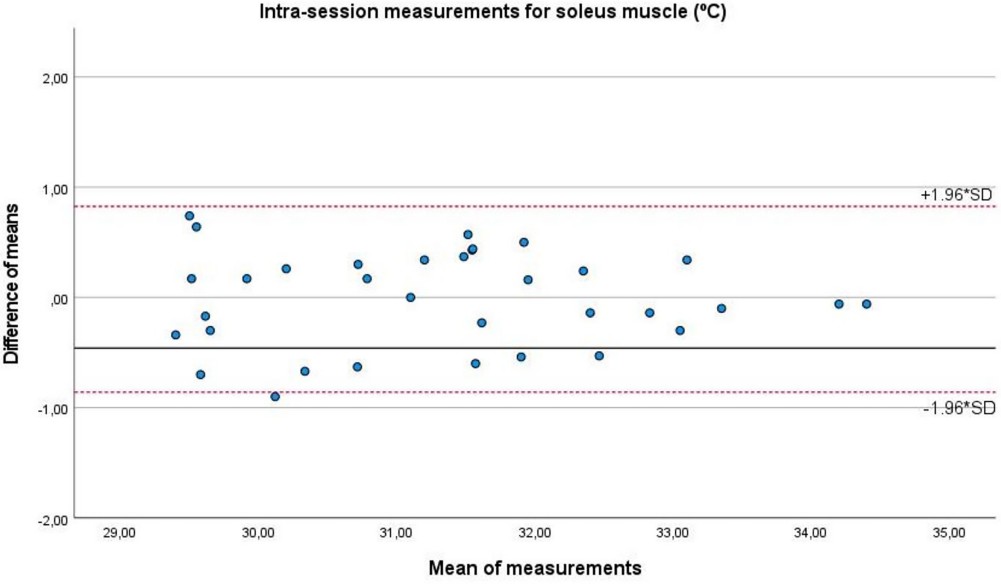

**Figure 3 Bland-Altman plots agreement for intra-session measurements of the infrared thermography device designed for materials (Manual Infrared Camera PCE-TC 30) in the soleus muscle tissue of athletes.** Completed with the upper and lower limits of agreement (LoA).

Regarding intra-session measurements, bivariate correlations were excellent for the soleus ($r = 0.953$; $P < 0.001$), medial gastrocnemius ($\rho = 0.885$; $P < 0.001$), and lateral gastrocnemius ($r = 0.939$; $P < 0.001$).

In addition, Bland-Altman plots presented an adequate agreement for intra-session measurements of the infrared thermography device designed for materials (Manual Infrared Camera PCE-TC 30) in the soleus (Fig. 3), medial gastrocnemius (Fig. 4) and lateral gastrocnemius (Fig. 5) muscles of athletes, due to visual distributions of the difference means for each pair of measurements at Y axis concerning the mean for each pair of measurements at X-axis did not present any systematic measurement error and most thermographic measurements were between the upper and lower LoA. In addition to

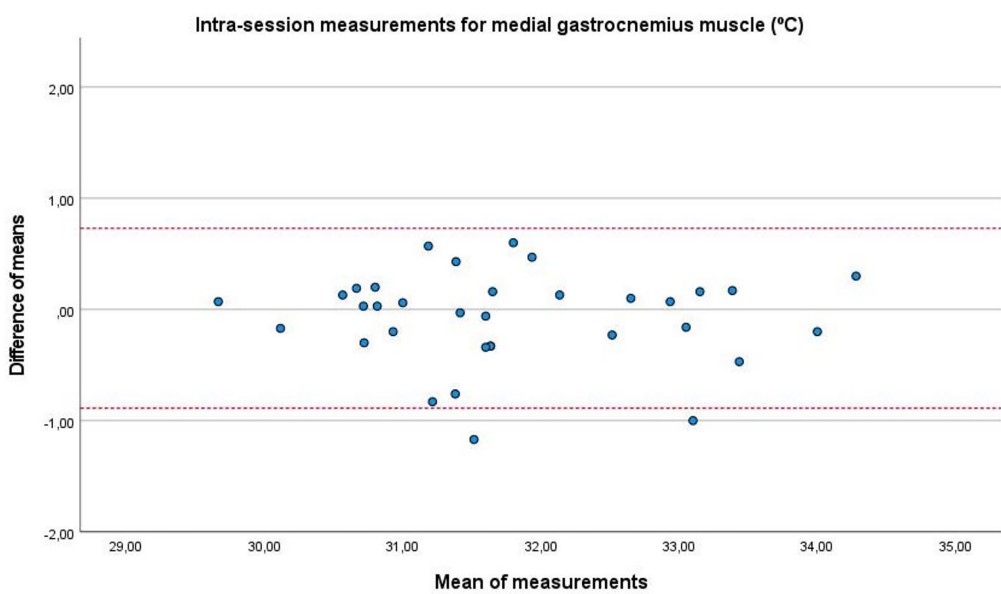

**Figure 4 Bland-Altman plots agreement for intra-session measurements of the infrared thermography device designed for materials (Manual Infrared Camera PCE-TC 30) in the medial gastrocnemius muscle tissue of athletes.** Completed with the upper and lower limits of agreement (LoA).             

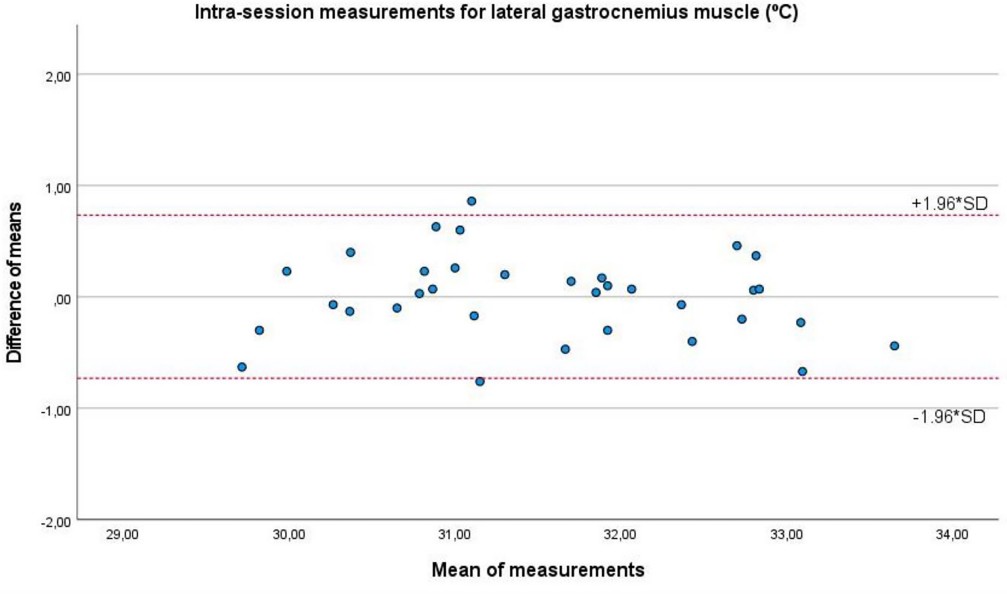

**Figure 5 Bland-Altman plots agreement for intra-session measurements of the infrared thermography device designed for materials (Manual Infrared Camera PCE-TC 30) in athletes' lateral gastrocnemius muscle tissue.** Completed with the upper and lower limits of agreement (LoA).             

**Table 3 Inter-session reliability and repeatability of an infrared thermography device designed for materials (Manual Infrared Camera PCE-TC 30) in the triceps surae muscle tissue of athletes.**

| Infrared thermography (°C) | Baseline measurements (95% CI) | After 48 h measurements (95% CI) | $ICC_{(1,2)}$ (95% CI) | SEM | MDC | *P*-value[*] |
|---|---|---|---|---|---|---|
| Soleus | 31.31 ± 1.41 [30.82–31.81] | 31.24 ± 1.36 [30.76–31.71] | 0.974 [0.948–0.987] | 0.222 | 0.615 | 0.338[‡] |
| Medial gastrocnemius | 31.74 ± 1.12 [31.34–32.13] | 31.77 ± 1.11 [31.38–32.16] | 0.956 [0.913–0.978] | 0.232 | 0.643 | 0.682[‡] |
| Lateral gastrocnemius | 31.55 ± 1.03 [31.18–31.91] | 31.52 ± 1.08 [31.14–31.90] | 0.968 [0.937–0.984] | 0.187 | 0.518 | 0.719[‡] |

Notes:
[‡] Mean ± standard deviation and Student *t* test for paired samples were used.
[*] *P* < 0.05 was considered as statistically significant for a 95% CI.
CI, confidence interval; ICC, intraclass correlation coefficient; MDC, minimum detectable change; SEM, standard error of measurement.

Bland-Altman plots, linear regression models did not show any statistical significance for soleus ($R^2 = 0.000$; $\beta = 0.003$; $F_{1,32} = 0.003$; $P = 0.954$), medial gastrocnemius ($R^2 = 0.002$; $\beta = -0.017$; $F_{1,32} = 0.065$; $P = 0.800$), and lateral gastrocnemius ($R^2 = 0.019$; $\beta = -0.050$; $F_{1,32} = 0.626$; $P = 0.435$) intra-session measurements.

### Inter-session reliability and repeatability

According to Table 3, inter-session measurements of the infrared thermography device designed for materials (Manual Infrared Camera PCE-TC 30) in the triceps surae muscle tissue of athletes presented excellent reliability ($ICC_{(1,2)} = 0.956$–$0.974$), measurement errors (SEM = 0.187–0.232 °C; MDC = 0.518–0.643 °C), and did not present any statistically significant systematic error of measurements ($P > 0.05$).

Considering inter-session measurements, bivariate correlations were also excellent for the soleus ($r = 0.949$; $P < 0.001$), medial gastrocnemius ($r = 0.914$; $P < 0.001$), and lateral gastrocnemius ($r = 0.938$; $P < 0.001$).

Lastly, Bland-Altman plots showed an adequate agreement for inter-session measurements of the infrared thermography device designed for materials (Manual Infrared Camera PCE-TC 30) in the soleus (Fig. 6), medial gastrocnemius (Fig. 7), and lateral gastrocnemius (Fig. 8) muscles of athletes, since visual distributions of the difference means for each pair of measurements at the Y axis concerning the mean for each pair of measurements at the X axis did not present any systematic measurement error, and most thermographic measurements were between the upper and lower LoA.

In conjunction with the Bland-Altman plots, linear regression models did not show any statistical significance for soleus ($R^2 = 0.014$; $\beta = 0.039$; $F_{1,32} = 0.463$; $P = 0.501$), medial gastrocnemius ($R^2 = 0.000$; $\beta = -0.009$; $F_{1,32} = 0.014$; $P = 0.907$), and lateral gastrocnemius ($R^2 = 0.019$; $\beta = -0.050$; $F_{1,32} = 0.621$; $P = 0.436$) inter-session measurements.

## DISCUSSION

The proposed procedure within the Manual Infrared Camera PCE-TC 30 thermography device presented excellent intra- and inter-session reliability and repeatability with an adequate agreement avoiding systematic errors of measurement to measure skin temperature of soleus, medial and lateral gastrocnemius muscles. Although this tool was initially designed to assess the superficial temperature of materials (*De Villoria et al., 2011*; *Pérez-Urrestarazu et al., 2014*), it could be a less expensive device to promote triceps surae
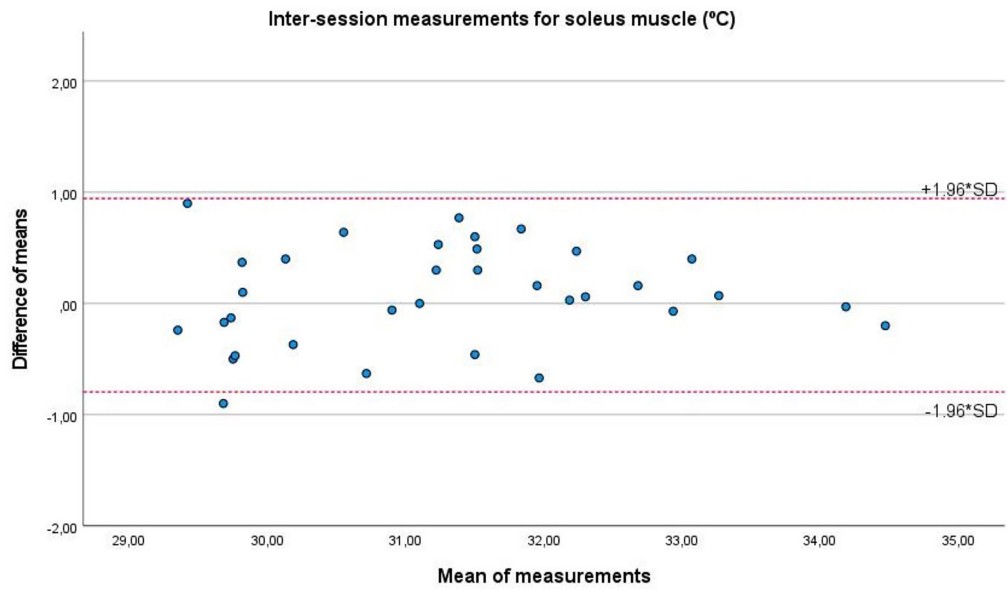

**Figure 6 Bland-Altman plots agreement for inter-session measurements of the infrared thermography device designed for materials (Manual Infrared Camera PCE-TC 30) in the soleus muscle tissue of athletes.** Completed with the upper and lower limits of agreement (LoA).

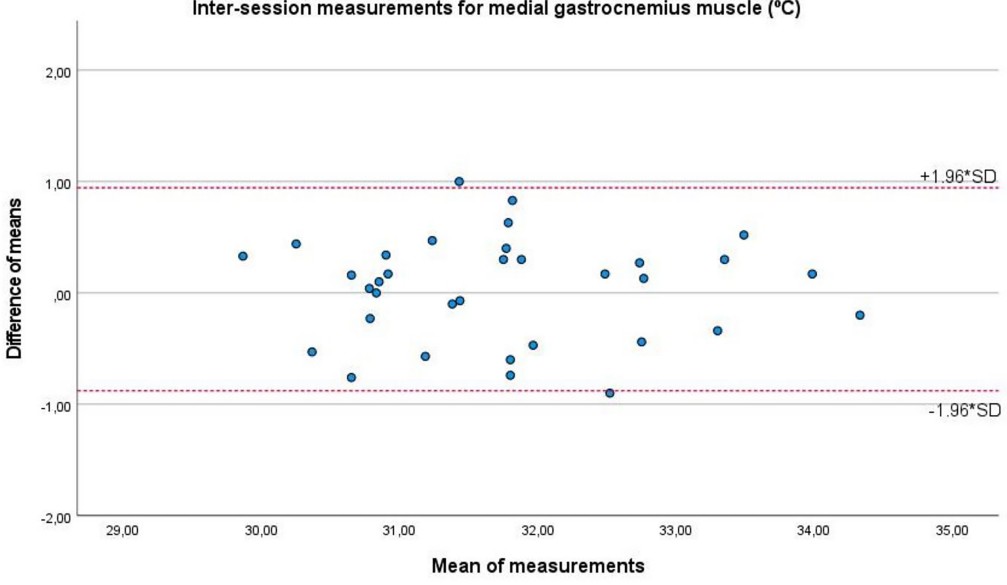

**Figure 7 Bland-Altman plots agreement for inter-session measurements of the infrared thermography device designed for materials (Manual Infrared Camera PCE-TC 30) in the medial gastrocnemius muscle tissue of athletes.** Completed with the upper and lower limits of agreement (LoA).

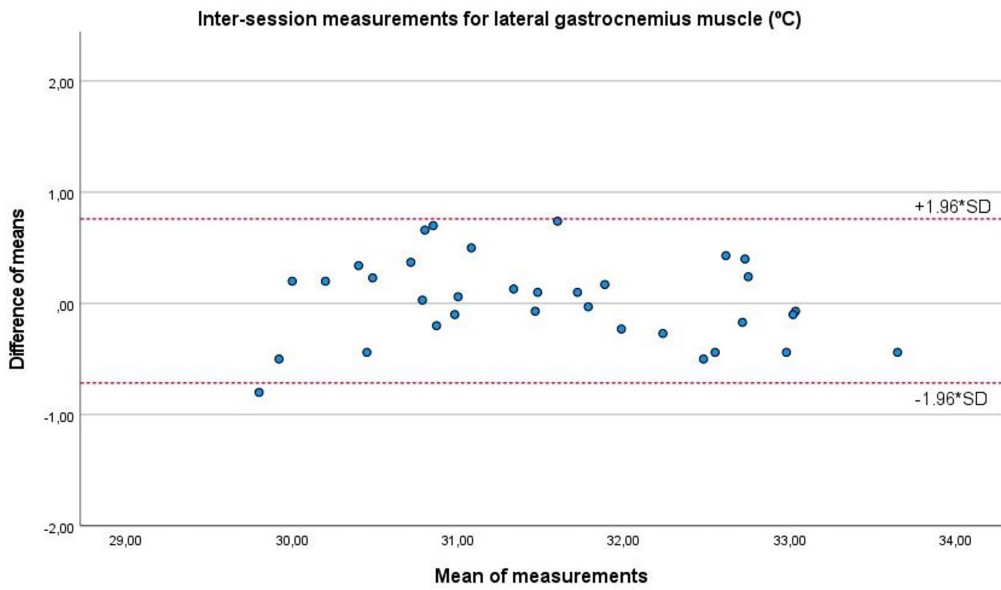

**Figure 8 Bland-Altman plots agreement for inter-session measurements of the infrared thermography device designed for materials (Manual Infrared Camera PCE-TC 30) in athletes' lateral gastrocnemius muscle tissue.** Completed with the upper and lower limits of agreement (LoA)

muscle prevention and recovery in athletes (*Rodríguez-Sanz et al., 2017*, *2018*; *Rodriguez et al., 2018*; *Benito-de-Pedro et al., 2019*). In addition, the PCE-TC 30 thermal camera may be employed to assess the thermal behavior of fencing uniforms in athletes (*Lamberti et al., 2020*). Our study supports that this device may be directly used to determine superficial human muscle tissue temperature in the triceps surae of athletes with adequate reliability and repeatability (*Rodríguez-Sanz et al., 2017*, *2018*; *Rodriguez et al., 2018*; *Benito-de-Pedro et al., 2019*; *Requena-Bueno et al., 2020*).

The PCE-TC 30 thermography device showed similar excellent reliability (ICC > 0.8) compared to the infrared ThermoHuman® device (*Fernandez-Cuevas et al., 2017*; *Requena-Bueno et al., 2020*), the digital infrared camera IRTIS-2000® (*Zaproudina et al., 2008*), and Thermofocus® thermal imaging device (*Petrova et al., 2018*), which were specifically designed for human tissue temperature measurements in lower limbs. Indeed, different ThermoHuman® devices showed an excellent intra-session (ICC = 0.99; LoA = 0.0 ± 0.4–0.1 ± 0.4 °C) and inter-session (ICC = 0.92; LoA = 0.1 ± 0.4–0.1 ± 0.5 °C) reliability before and after running, respectively, detailing small differences effect sizes (Cohen's $d$ < 0.4) for foot skin temperature (*Requena-Bueno et al., 2020*).

Specifically, the digital infrared camera IRTIS-2000® applied in the calf region showed adequate intra-session (ICC = 0.84) and inter-session (ICC = 0.66) reliability with similar temperature mean ± SD values for anterior (31.2 °C ± 0.6 °C), posterior (30.8 °C ± 0.6 °C), and lateral (31.3 °C ± 0.6 °C) calf regions (*Zaproudina et al., 2008*). Finally, the Thermofocus® thermal imaging device presented almost perfect intra-session agreement

in different foot regions (ICC = 0.94–0.97), showing non-significant replication interactions (*P*-values = 0.23–0.84) (*Petrova et al., 2018*). Despite the use of such modern thermal imaging systems with better definition resolutions could provide more reliable measurements and improve the quality of the thermograms obtained, the reliability and measurement errors provided by the Manual Infrared Camera PCE-TC 30 thermography may be enough to detect temperature differences linked to clinical musculoskeletal changes (*Côrte et al., 2019*).

Athletes may be exposed to physical stress under training loads and competitions with overload reactions which could cause blood flow changes affecting skin temperature (*Merla et al., 2010*). Infrared thermography may not display anatomical abnormalities, although functional changes may be shown and linked to skin temperature control (*Merla et al., 2010*; *Ring & Ammer, 2012*). Infrared thermography use was proposed as a complementary tool to apply preventive measures, such as cryotherapy, physiotherapy, training load reduction, and massage or recovery boot use, to avoid muscle conditions in professional soccer players. The asymmetry reference values range from 0.5 °C to 1 °C between both right and left lower limbs was proposed to initiate this preventive protocol, which reduced up to 63% muscle injuries in a professional soccer season by thermographic monitoring (*Côrte et al., 2019*). Thus, the Manual Infrared Camera PCE-TC 30 thermography device could be used to determine these cut-off values due to the MDC values varied from 0.515 °C to 0.587 °C and from 0.518 °C to 0.643 °C for intra- and inter-session evaluations, respectively.

Our research group carried out a prior thermographic study addressing the thermal skin evaluation of the triceps surae muscles. This study showed that skin temperature after running was deeply linked to electromyography, which may indirectly reflect triceps surae muscle activity. Although the Manual Infrared Camera PCE-TC 30 thermography device has not yet been correlated with electromyography values, thermal values could be related to muscle activity in the triceps surae muscles of athletes using this tool according to a similar study using the FLIR/SC3000/QWIP Thermacan-infrared thermal device (*Rodriguez-Sanz et al., 2019*).

Although some less expensive commercially-available thermal cameras could be suitable for skin temperature assessment, employing the PCE-TC 30 camera to assess triceps surae muscle temperature provided reliable and repeatable measures with MDC cut-off values useful to determine preventive protocols for muscle injuries (*Côrte et al., 2019*).

## Future studies

Further studies should be designed as randomized clinical trials to determine if these asymmetries reference cut-off values could prevent triceps surae muscle injuries (*Côrte et al., 2019*). According to prior studies (*Hiemstra et al., 2007*; *Chung et al., 2015*), the uninvolved normal side after injury may often be not normal, *i.e.*, presenting temperature values different from healthy subjects, and cut-off values should also be detailed in the future muscle recovery studies. In addition, thermographic measurement of the triceps surae with this device should be analyzed by intra- and inter-rater reliability determining

SEM and MDC values and correlated with the other high-end infrared devices as possible gold standards such as ThermoHuman® (*Fernandez-Cuevas et al., 2017*; *Requena-Bueno et al., 2020*), IRTIS-2000® (*Zaproudina et al., 2008*) and Thermofocus® (*Petrova et al., 2018*) tools. Furthermore, correlations with electromyography measurements of the triceps surae muscle activity should be carried out (*Rodriguez-Sanz et al., 2019*). Lastly, the intramuscular temperature should also be correlated with this device to determine concurrent validity concerning a gold standard (*Burnham, McKinley & Vincent, 2006*).

## Limitations

Various limitations should be considered for the use of this thermographic device. First, the MDC was superior to the lower limbs asymmetry reference range from 0.3 °C to 0.4 °C proposed as a cut-off for following-up before a preventive protocol (*Côrte et al., 2019*) and therefore, this device should not be used for values lower than 0.5 °C–0.6 °C.

Second, the concurrent validity of this device has not been performed for human tissue temperature, and this validity should be assessed in the future (*Burnham, McKinley & Vincent, 2006*).

Third, our sample only comprised healthy athletes, and further studies should analyze skin temperature with this tool over injured muscle tissue (*Alburquerque Santana et al., 2022*).

Fourth, our sample size calculation was accurately detailed to determine a moderate bivariate correlation between measurements, but our sample size was low to achieve the actual power to perform comparisons classifying groups depending on sport category, BMI, and other valued characteristics. In addition, despite statistical analyses were carried out according to our prior sample size calculation model to detail bivariate comparisons for intra- and inter-session measurements, future nested type studies should be designed as nested statistical models such as analyses of variance (ANOVA) to determine more accurate temperature comparisons.

Finally, in spite of the device was calibrated according to the manufacturer, the fact that the PCE-TC 30 thermography device was not designed for skin temperature assessment could affect the repeatability of the measurement to a lesser extent than its accuracy. Nevertheless, a comparison between the temperature assessed by a validated thermal camera and the PCE-TC 30 was not reported. Future studies should evaluate its accuracy due to a possible wrong estimation of the absolute skin temperature. Procedures for thermographic assessment in sports and exercise sciences have been reviewed by *Moreira et al. (2017)* in a consensus statement of the experts in the field. However, our reliability study aimed to standardize the proposed procedure and contributed to improving the methods behind measures.

## CONCLUSION

The proposed procedure within the Manual Infrared Camera PCE-TC 30 thermography device designed initially to measure the superficial temperature of materials presented excellent intra- and inter-session reliability and repeatability to measure skin temperature in the triceps surae muscle tissue of athletes. Future studies should consider the

measurement errors of this procedure to measure the skin temperature of soleus, medial, and lateral gastrocnemius muscles to promote triceps surae muscle prevention and recovery in athletes.

### Funding

César Calvo-Lobo, David Rodríguez-Sanz and Ricardo Becerro-de-Bengoa-Vallejo declared that this work has been supported by a Contract 83 between the Complutense University and PCE Ibérica S.L. entitle "Fiabilidad sobre el dispositivo de termografía Manual Infrared camera PCE-TC 30 en humanos" (Reference number: 6-2020). The funders had no role in study design, data collection and analysis, decision to publish, or preparation of the manuscript.

### Grant Disclosures

The following grant information was disclosed by the authors:
Complutense University and PCE Ibérica S.L: Ref. No. 6-2020.

### Competing Interests

César Calvo-Lobo, David Rodríguez-Sanz and Ricardo Becerro-de-Bengoa-Vallejo declare conflict of interest due to the Contract 83 between the Complutense University and PCE Ibérica S.L. (Reference number: 6-2020) was specifically carried out to study the reliability of the Manual Infrared camera PCE-TC 30 thermography device.

### Author Contributions

- Cesar Calvo-Lobo conceived and designed the experiments, performed the experiments, analyzed the data, prepared figures and/or tables, authored or reviewed drafts of the article, and approved the final draft.
- Marta San-Antolín conceived and designed the experiments, performed the experiments, analyzed the data, prepared figures and/or tables, authored or reviewed drafts of the article, and approved the final draft.
- Daniel García-García conceived and designed the experiments, performed the experiments, analyzed the data, prepared figures and/or tables, authored or reviewed drafts of the article, and approved the final draft.
- Ricardo Becerro-de-Bengoa-Vallejo conceived and designed the experiments, performed the experiments, analyzed the data, prepared figures and/or tables, authored or reviewed drafts of the article, and approved the final draft.
- Marta Elena Losa-Iglesias conceived and designed the experiments, performed the experiments, analyzed the data, prepared figures and/or tables, authored or reviewed drafts of the article, and approved the final draft.
- Julia Cosín-Matamoros performed the experiments, authored or reviewed drafts of the article, and approved the final draft.
- Israel Casado-Hernández performed the experiments, authored or reviewed drafts of the article, and approved the final draft.

- Eva María Martínez-Jiménez performed the experiments, authored or reviewed drafts of the article, and approved the final draft.
- Victoria Mazoteras-Pardo performed the experiments, authored or reviewed drafts of the article, and approved the final draft.
- David Rodríguez-Sanz conceived and designed the experiments, performed the experiments, analyzed the data, prepared figures and/or tables, authored or reviewed drafts of the article, and approved the final draft.

## Human Ethics

The following information was supplied relating to ethical approvals (*i.e.*, approving body and any reference numbers):

The Clinical intervention ethics committee of the Hospital Clínico San Carlos, Madrid (Spain) approved the study (20/021-E).

## Data Availability

The raw data are available in the Supplemental Files.

## Supplemental Information

Supplemental information for this article can be found online at http://dx.doi.org/10.7717/peerj.15011#supplemental-information.

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
