# Peer review of "Intra- and inter-session reliability and repeatability of an infrared thermography device designed for materials to measure skin temperature of the triceps surae muscle tissue of athletes"

_PeerJ, doi:10.7717/peerj.15011_

## Round 0.1 · original submission · Major Revisions

Dear Author,

Please carefully revise your paper based on my and the reviewers' comments.
You must get your manuscript edited by a professional in scientific writing. Please verify that the references are appropriate.

Regards

Reviewer 1 ·

Basic reporting

The paper investigates the intra- and inter-session reliability and repeatability of manual infrared camera PCE-TC 30, which is designed for materials, to assess skin temperature of the triceps surae muscle tissue in athletes. The paper is well written, reports an appropriate investigation of the literature and the quality of the results presentation is good.

Experimental design

The Authors should specify whether the participants underwent to a period of acclimation, fundamental to perform repeatible thermal measurements.

Validity of the findings

1) The Authors should explain why it is hypothesized that a thermal camera designed for materials does not provide repeatable measurements for skin temperature. In fact, maybe the fact that it is not calibrated for skin temperature assessment does not affect the repeatability of the measurement but its accuracy. However, in the paper, a comparison between the temperature assessed by a validated thermal camera and the PCE-TC 30 is not reported. Please, justify your hypothesis and describe possible limitations due
to a possible wrong estimation of the absolute skin temperature.

2) The PCE-TC 30 thermal camera was already employed to assess skin temperature to evaluate the thermal behaviour of fencers.

Lamberti, G., Leccese, F., Salvadori, G., & Fantozzi, F. (2020). Effect of Exercise on Athletes Performing in Fencing Uniforms: Methodology and Preliminary Results of the Use of Infrared Thermography to Detect the Thermal Behaviour of Fencers. Applied Sciences, 10(9), 3296.

Please, highlight the benefit to literature of your findings.

Additional comments

The Authors should also stress the advantage of employing the PCE-TC 30 camera to assess triceps surae muscle temperature with respect to another thermal camera. In fact, although it is cheaper with respect to several thermal imaging device, some less expensive thermal cameras suitable for skin temperature assessment are commercially available.

Reviewer 2 ·

Basic reporting

This study investigated the intra- and inter-session reliability of a specific device designed for materials to measure skin temperature of the triceps surae in an athletes sample. This a methodological study, aimed to assess reliability in the field of skin temperature assessment. The manuscript is well written, easy-to-read, and methodologically correct. I would like to congratulate the authors for their work. I have only some suggestions that I hope may be useful to improve the overall quality of the manuscript.
• There is previous literature on the skin temperature assessment by thermography on the specific regions of calf in athletes and its reliability. I strongly suggest to cite some papers to reinforce lines 55-60.
• The Authors should explain the rationale behind this study. Is the device tested not reliable for materials? As reliable for materials, this should be reliable also for skin temperature. Rather, this should be a problem of accuracy…
• On the same line. Is it possible that the Authors are investigating the reliability of the procedures? Instead of the thermal camera? Procedures for thermographic assessment in sport and exercise sciences have been reviewed by Moreira et al. (2017) in a consensus statement of the experts in the field. However, reliability studies aimed at standardize procedures are important and contribute to improve the methods behind measures. In this sense, consider to shift the focus of the study from the reliability of the devices, to the reliability of procedures.
• I suggest to include a figure explaining the experimental procedures, thus allowing a reader a better comprehension of the design employed.

Experimental design

no comment

Validity of the findings

no comment

---

## Round 0.2 · Minor Revisions

Dear authors:

I appreciate you paying attention to the reviewers' comments. I'm pleased to suggest that your article is almost ready to be published. The Section Editor and I have found a few minor errors that must be fixed before publication.

---

## Round 0.3 · accepted · Accept

Thanks a lot for promptly attending to the corrections recommended by the section editor. I am happy to recommend your article for publication.